# Stimulation of THP-1 Macrophages with LPS Increased the Production of Osteopontin-Encapsulating Exosome

**DOI:** 10.3390/ijms21228490

**Published:** 2020-11-11

**Authors:** Gaowa Bai, Takashi Matsuba, Toshiro Niki, Toshio Hattori

**Affiliations:** 1Department of Health Science and Social Welfare, Kibi International University, Takahashi 716-8508, Japan; gaowabai@kiui.ac.jp; 2Division of Bacteriology, Department of Microbiology and Immunology, Faculty of Medicine, Tottori University, Yonago, Tottori 683-8503, Japan; matsubat@tottori-u.ac.jp; 3Department of Immunology, Kagawa University, Kita-gun, Kagawa 7610793, Japan; niki@med.kagawa-u.ac.jp

**Keywords:** osteopontin, THP-1, exosome, LPS, Exo-Prep

## Abstract

Osteopontin (OPN) mediates bone remodeling and tissue debridement. The OPN protein is cleaved, but it is unclear how full-length (FL)-OPN or its cleaved form perform their biological activities in target cells. We, therefore, performed the molecular characterization of OPN in exosomes (Exo). The Exo were isolated from lipopolysaccharide (LPS)-stimulated phorbol 12-myristate 13-acetate (PMA)-differentiated THP-1 macrophages. The Exo were also isolated from PMA-differentiated THP-1 macrophages. The Exo were identified using the qNano multiple analyzer (diameter 59–315 nm) and western blotting with a CD9 antibody. LPS-stimulated cells produced more particles than non-stimulated cells. The presence of the FL or the cleaved form of OPN was confirmed using western blot analysis. A mixture of FL and cleaved OPN was also measured using an ELISA system (Ud-OPN) and their presence in the Exo was confirmed. Ud/FL ratios became low after LPS stimulation, indicating the enhanced encapsulation of FL-OPN in the Exo by LPS. These findings suggest that LPS stimulation of human macrophages facilitates the synthesis of FL-OPN, which is cleaved in cells or the Exo after release. These findings indicate that Exo is a suitable vehicle to transfer OPN to the target cells.

## 1. Introduction

Osteopontin (OPN) is a calcium-binding glycophosphoprotein that was originally isolated in the bone; accordingly, OPN mediates bone remodeling and tissue debridement [1,2]. OPN is produced by macrophages and has been shown to mediate immunity, inflammation, tumor progression, and cell viability [3,4], as well as cell adhesion, proliferation, invasion, and apoptosis during tissue fibrosis [5]. It has been previously suggested to mediate cancer progression via its expression in cancer and lymphoma cells [6]. In support of this, OPN expression is associated with the performance status, the total number of involved lesions, lactic dehydrogenase levels, and lymphocyte count, and thus predicts poor survival in adult T cell leukemia (ATL) [3]. OPN expression is also enhanced by hypoxic conditions [7,8], and high OPN levels correlate with plasma Hsp70 levels and are associated with a reduced overall survival rate in non-metastasized non-small-cell lung cancer [9].

Proteolytic OPN cleavage by thrombin (between Arg168 and Ser169) generates a functional *N*-terminal OPN fragment (truncated OPN, *N*-half) that includes an integrin-binding site which functions to promote neutrophil survival, thereby contributing to the pathogenesis of both ATL and anti-type II collagen antibody-induced arthritis [10,11]. We previously reported the up- and downregulation of full-length (FL)-OPN in the plasma of patients with acute and recovery-phase dengue virus infections, respectively. Conversely, there was an upregulation of a truncated form of OPN (*N*-half) in patients in the recovery phase, suggesting that FL and *N*-half OPN respond differently to disease conditions [12]. Furthermore, the titers of OPN measured using another ELISA (Ud-OPN) showed the rapid increase in the acute phase like FL-OPN but did not decrease during the recovery phase and was proposed to measure a mixture of FL and a cleaved form of OPN. The ELISA is named as Undefined (Ud)-OPN, because the epitope by ELISA is undefined [12] and reacts strongly with OPN in the supernatant of phorbol 12-myristate 13-acetate (PMA)-stimulated THP-1 macrophage [13]. We have previously shown that plasma Ud-OPN levels are elevated in patients with tuberculosis (TB), and only Ud-OPN levels but not those of FL-OPN are positively correlated with the numbers of TB-specific interferon (IFN)-γ-secreting memory T cells and neutrophils, respectively. Together, these findings suggest that OPN promotes granuloma formation by mediating the migration of lymphocytes and neutrophils [14].

As described above, Ud-OPN is detected in the supernatants of PMA-differentiated THP-1 macrophages, indicating that other proteins may be involved in the generation of Ud-OPN. Matrix metalloproteinase 9 (MMP-9) can cleave OPN [15], is highly induced in PMA-differentiated THP-1 macrophage [16], and has been hypothesized to mediate OPN cleavage intracellularly [13]. Furthermore, the proteolytic cleavage of OPN by caspase-8 in cells and its cleavage product induces cell death via p53 [17]. Recently, many cell-secreted factors (e.g., cytokines and nucleic acids) have been shown to be transported to their targets by extracellular vesicles (Ev), including exosomes (Exo) (30–200 nm) [18]. For example, the Exo derived from dendritic cells were recently shown to contain OPN and MMP-9 and could enhance the migration of mesenchymal stem cells and stromal cells [19]. Moreover, intranasal injection of the Exo isolated from *Mycobacterium bovis* Bacillus Calmette Guérin (BCG)- and *M. tuberculosis*-infected macrophages stimulated TNF-α and IL-12 production as well as neutrophil and macrophage recruitment in the mouse lung [20]. PMA-differentiated THP-1 macrophages also have a role in the Exo-mediated transfer of IFN-γ-induced antiviral molecules [21]. The Exo isolated from the supernatants of lipopolysaccharide (LPS)-stimulated PMA-differentiated THP-1 macrophages enhanced the proliferation and activation of cells [22]. Therefore, we herein characterized the levels of OPN in the Exo isolated from LPS-stimulated PMA-differentiated THP-1 macrophages.

## 2. Results

### 2.1. LPS Stimulation of PMA-Differentiated THP-1 Macrophages Enhanced FL-OPN Synthesis

Two different ELISA systems were used: one which could detect FL-OPN and, the other, a mixture of FL-OPN and degraded OPN (Ud-OPN). Both ELISA systems showed a dramatic increase of OPN in the supernatants of PMA-differentiated THP-1 macrophages (Figure 1A,B). The LPS stimulation of PMA-differentiated THP-1 macrophages increased FL-OPN significantly, but Ud-OPN was not changed, although the constitutive production of Ud-OPN was observed even without LPS stimulation. These findings support that LPS stimulation may be a suitable stimulant for FL-OPN synthesis.

### 2.2. LPS Stimulation of PMA-Differentiated THP-1 Macrophages Produced More Exo

Tunable resistive pulse sensing (TRPS) showed that the diameter of the Exo isolated from the culture supernatants of the LPS-stimulated THP-1-differentiated macrophages ranged from 59–315 nm with a mode diameter of 78 nm. Exo from culture supernatants of non-stimulated THP-1-differentiated macrophages ranged from 97–467 nm with a mode diameter of 120 nm (Figure 2A). More particles (*p* < 0.0001) were found in the Exo from culture supernatants of LPS-stimulated THP-1 macrophages as compared with those from non-stimulated THP-1 macrophages (Figure 2B). The diameter was significantly smaller in LPS-stimulated cells (*p* < 0.0001) (Figure 2C).

### 2.3. Enhanced Encapsulation of FL-OPN in the Exo by LPS Stimulation

Optimization of ELISA conditions was performed as it was unclear whether the conventional sample diluent would be suitable for the quantitation of OPN in the Exo. In ELISA for the Exo, we tested whether Triton X-100 (1% *v*/*v*) is effective in improving OPN detection in the Exo. The addition of Triton X-100 did not affect the results when control antigens were used (data not shown). However, if Exo was used as an antigen source, the presence of Triton X-100 dramatically increased the amount of Ud-OPN detected (Figure 3).

OPNs in the Exo isolated from culture supernatants of PMA-differentiated THP-1 macrophages and LPS-stimulated THP-1 macrophages were measured (Table 1). In contrast to the culture supernatants, the amount of both FL-OPN and Ud-OPN were lower in the Exo isolated from the culture supernatant of PMA-differentiated THP-1 macrophages than those isolated from LPS-stimulated THP-1 macrophages. On the contrary, the Exo derived from the supernatants of LPS-stimulated PMA-differentiated THP-1 macrophages had increased FL-OPN compared to those from PMA-differentiated THP-1 macrophages. Furthermore, the supernatant isolated on day 2 contained the highest amount of FL-OPN. The amounts of Ud-OPN in the Exo isolated from LPS-stimulated (day 2) PMA-differentiated THP-1 macrophages were also higher compared to that of the PMA-differentiated THP-1 macrophages.

The ratios of OPN in the Exo and culture supernatants were calculated. The amounts of FL-OPN and Ud-OPN were calculated based on the manufacturer’s instructions. Here, 900 μL of Exo was obtained from 9 mL of the culture supernatants. Approximately 15–17% of FL-OPN and 16–19% of Ud-OPN in the culture supernatants were present in the Exo in the LPS-stimulated cultures, while less than 1% of FL-OPN and 2.6% Ud-OPN in the culture supernatants of the PMA-differentiated THP-1 macrophages were sequestered in the Exo. Furthermore, the ratios of Ud-OPN to FL-OPN were very high only in the Exo isolated from the culture supernatant of the PMA-differentiated THP-1 macrophages (Table 1).

### 2.4. Detection of Both FL and Cleaved OPN in the Exo from LPS-Stimulated THP-1 Macrophage

A comparative study of OPN in the culture supernatants and the Exo was performed using western blot analysis (Figure 4). The bands of OPN are seen in the Exo as well as in the culture supernatants. The intensities of the bands also roughly correlated with the amount measured using ELISA (Table 1). Cell lysates of PMA-stimulated THP-1 cells and the Exo from LPS-stimulated PMA-differentiated THP-1 macrophage showed clear strong bands corresponding to 75 kDa of FL-OPN and 53 kDa of its cleaved form (Figure 4A, lane 4,6) using O-17 antibody. The anti-CD9 antibody also gave a strong band at 24 kDa against the Exo and cell lysates (Figure 4B, lane 4 and 6).

### 2.5. Protease Inhibitors Did Not Show an Apparent Effect on OPN Processing and Transcription in LPS-Stimulated THP-1 Macrophage

The apparent effect of protease inhibitors on the profiles of western blot of cell lysates of LPS-simulated PMA-differentiated THP-1 macrophage using O-17 antibody was not observed, indicating exogenous addition of the protease inhibitors against MMP-9 [23] and caspase-8 [24] did not affect both FL and cleaved products of OPN synthesis (Appendix A). The levels of Ud-OPN in the culture supernatants also did not change (data not shown). In RT-PCR assay only one band of OPNa corresponding to FL-OPN was observed in various stimulated cells, but other bands corresponding to splicing variants in RNA transcripts were not observed (Appendix A).

## 3. Discussion

Both sterile and non-sterile inflammatory conditions induce monocytes to generate Ev using THP-1 cells [20]. THP-1 cells are known to differentiate into macrophages after at least 48 h of incubation with PMA [21]. PMA-differentiated THP-1 macrophages also secrete Exo after the treatment of LPS [19]. OPN enhances the Th1-mediated inflammatory response and plays a key role in apoptosis. High OPN expression induced by interferon regulatory factor 8 downregulation in tumor cells was found to suppress the activation of CD8 cells by binding to their CD44 receptors. Therefore it was suggested that OPN may also act as an immune checkpoint molecule [25]. OPN is known to be susceptible to cleavage by different proteases like thrombin and metalloproteinases [15]. PMA-differentiated THP-1 macrophages synthesize both full length and cleaved forms of OPN, indicating intracellular cleavage of OPN in THP-1 cells [13]. Intracellular cleavage of OPN by caspase 8 has been shown [17]. We chose LPS-stimulated PMA-differentiated THP-1 macrophages as a source of Exo. The presence of Exo, whose diameter ranged from 59–315 nm, was confirmed using TRPS in our preparations. It is of note that the Exo derived from LPS-stimulated PMA-differentiated THP-1 macrophages were significantly smaller than those from non-stimulated cells and may indicate the fragmentation of the Exo by stimulation. The presence of CD9, which identifies the Exo [26,27], was also confirmed. There are currently no reports if a cleaved form of OPN is present in the Exo. Western blot analyses using the O-17 antibody, which recognizes the *N*-terminal OPN, gave intense bands corresponding to both FL and cleaved forms of OPN. The increase in detection sensitivity of the cleaved form in the supernatant of the LPS-stimulated PMA-differentiated THP-1 macrophages (Figure 4A, lane 2) was due to the usage of sodium dodecyl sulfate (SDS)/urea for the sample treatment, which is different from SDS/2-mercaptoethanol (data not shown). ELISA showed that the Exo isolated from the culture supernatants in LPS-stimulated PMA-differentiated THP-1 cells contained FL-OPN and Ud-OPN and their contents were 16–19% of those of the culture supernatants. The presence of the cleaved form, as well as FL-OPN in the Exo and culture supernatants, was confirmed via western blotting for the first time. In the current study, we were able to detect the band corresponding to the cleaved form without the need for any artificial digestion. MMP-9 is the most highly induced protein in the PMA-treated THP-1 cells, and its activities may be responsible for its cleavage [16]. Furthermore, caspase 8 is also reported to be present in PMA-differentiated THP-1 cells [28].

The increase in the ratios of protein levels in the Exo derived from the LPS-stimulated PMA-differentiated THP-1 macrophages compared to those from the PMA-differentiated THP-1 macrophages indicates that the FL-OPN and the cleaved form of OPN could be encapsulated in the Exo after LPS stimulation. The lower ratios of Ud-/FL-OPN and the highest amounts of FL-OPN in the Exo from the LPS-stimulated PMA-differentiated THP-1 macrophages may be caused by the enhanced encapsulation of FL-OPN by LPS (Table 1), and the increased number of Exo in LPS-stimulated samples also indicates that the release of the Exo could also be enhanced by LPS (Figure 2). In contrast, FL-OPN in the Exo from PMA-differentiated THP-1 macrophages was very low, though the highest amount of FL-OPN is detected in their culture supernatants. The highest levels of Ud-OPN production in PMA-differentiated THP-1 macrophages may be caused by the presence of high amounts of MMP-9 [16] and/or caspase 8 [28] in these cells are also known. However, no reduction of cleaved products of OPN was observed with the administration of their inhibitors to the cultured stimulated THP-1 cells.

The Exo are cell-secreted, membrane-bound particles that are related to the endosomal pathway; an inward blebbing of the endosomal membrane produces intraluminal vesicles that are actively exocytosed as the Exo [29]. Proteins commonly identified in microvesicles include cytoskeletal proteins, heat shock proteins, integrins, and proteins containing post-translational modifications, such as glycosylation and phosphorylation [30]. OPN is a well-known glycoprotein that was identified in the Exo in this study.

In Exo derived from dendritic cells, interleukin 5, FL-OPN, fibroblast growth factor 7, and monocyte chemotactic protein-1 appeared highly enriched (fold-change > 2) and the active form of MMP-9 was indeed inside the vesicle [19]. Therefore, it is not clear if the cleavage occurs inside the cells or in the Exo. DC-derived microvesicles are a chemoattractant for mesenchymal stromal/stem cells [19]. The Exo released from LPS-activated THP-1 macrophages could enhance the proliferation and activation of hepatic satellite cells (HSC), which suggests an important role for the Exo secreted by infiltrated macrophages in regulating HSC survival and/or activation, thus accelerating the development of fibrosis [22]. These studies identify a previously unknown function of the Exo in promoting intercellular communication during an immune response to intracellular pathogens. It was hypothesized that the extracellular release of Exo-containing pathogen-associated molecular patterns is an important mechanism of immune surveillance [20]. The Exo derived from bacterial-infected macrophages carries bacterial coat components that stimulate bystander macrophages and neutrophils to secrete proinflammatory mediators [31]. The Exo isolated from LPS-stimulated PMA-differentiated THP-1 macrophages to evoke a pro-inflammatory profile in spleen cells of healthy mice through the induction of cytokines such as tumor necrosis factor-alpha (TNF-α), chemokine (C-C motif) ligand 5 (CCL5, also known as RANTES), and interleukin 1 beta (IL-1β) [32]. We previously reported that plasma OPN levels in *M. tuberculosis*-infected individuals are associated with granuloma formation through a positive association with neutrophil numbers and negative association with memory T cell numbers [17]. It is possible these effects were mediated by the Exo in the plasma released from activated macrophages in vivo. Further research on the biological functions of the Exo in patients’ serum would clarify these possibilities.

## 4. Materials and Methods

### 4.1. Cell Lines and Culture

Human THP-1 cells were obtained from the cell resource center for biomedical research, IDAC, Tohoku University (Sendai, Japan) [33]. All cells were maintained at 37 °C, 5% CO_2_, in a humidified atmosphere in Roswell Park Memorial Institute (RPMI) 1640 medium (Thermo Fisher Scientific, Waltham, MA, USA) that was supplemented with 10% heat-inactivated fetal bovine serum (FBS) (Thermo Fisher Scientific) and 1% kanamycin.

### 4.2. Exosome Isolation 

Two and a half million THP-1 cells were cultured in tissue culture dish 100 (TPP; Sigma-Aldrich Co. LLC. Tokyo, Japan) at 37 °C in a humidified atmosphere with 5% CO_2_ in Roswell Park Memorial Institute (RPMI) 1640 medium (Wako Pure Chemical Industries Ltd., Osaka, Japan) that was supplemented with 10% exosome-free FBS (System Biosciences, Palo Alto, CA, USA) and 1% kanamycin. They were then treated with PMA (10 ng/mL) for 48 h and the medium was replaced with a fresh medium one day later. Cells were then stimulated with 10 ng/mL of LPS-EB Ultrapure (Invivogen, San Diego, CA, USA) for 24 or 48 h and the supernatants were obtained via three centrifugation steps (300× *g* for 10 min, 1200× *g* for 20 min, and 10,000× *g* for 30 min) to remove cell debris and microvesicles (hence, the pellet was removed after each step). Supernatant cells stimulated with and without PMA were used as controls. Exosomes were isolated using the EXO-Prep exosome precipitation kit (HansaBioMed Life Sciences, Tallinn, Estonia) according to the manufacturer’s instructions. Briefly, 9 mL of the supernatant was thoroughly mixed with 9 mL of EXO-Prep exosome precipitation solution, and the resulting solution was incubated for 1 h at 4 °C, before being centrifuged at 10,000× *g* for 20 min. The generated (exosome-containing) pellet was suspended in 900 µL of PBS.

### 4.3. Tunable Resistive Pulse Sensing

Tunable resistive pulse sensing (TRPS) by qNano (Izon, Cambridge, MA, USA) was used to measure the size distribution and concentration of particles in isolated exosome fractions. To prevent protein binding to the pore, an Izon reagent kit was used, and PBS was added to each sample. An aliquot of exosomes from each sample or calibration particles included in the reagent kit (1:1, 000 EV, Izon) were placed in the Nanopore (NP100, Izon). The calibration particles were measured directly after the experimental sample under identical conditions. The sizes and concentrations of particles were determined using software provided by Izon (version 3.3).

### 4.4. ELISA

To identify FL-OPN, an ELISA kit (JP27158, IBL, Gunma, Japan) was used [3]. In the FL-OPN kit, O-17, a polyclonal rabbit antibody specific to Iso17-Gln31, was used as a capture antibody, and the mouse monoclonal antibody specific to Lys166–Glu187, 10A16, served as the detector antibody. These combinations of antibodies allowed us to detect FL-OPN [13]. Ud-OPN in the culture supernatants was detected using a Human OPN DuoSet ELISA Development System Kit (R&D Systems, Minneapolis, MN, USA) [12]. The proprietary capture monoclonal antibody and the detection of polyclonal antibodies in this ELISA kit were both generated against recombinant human OPN (NS0-derived; amino acids, Ile17-Asn300); the epitopes for these antibodies were not disclosed. To expose the protein in the Exo, samples containing the Exo and control antigens were suspended with PBS containing Triton X-100 (1%) for 30 min at room temperature. 

### 4.5. Western Blotting

Twenty-five million THP-1 cells differentiated with PMA in a 25 cm^2^ flask were washed and lysed on ice for 15 min using 1 mL of lysis buffer. The cell lysates were then processed using centrifugation at 14,000× *g* for 15 min at 4 °C as described [13]. The THP-1 culture supernatants were concentrated using Amicon^®^ Ultra centrifugal filters (Millipore). Protein concentrations were then determined using the Bradford method (Takara Bio Inc., Shiga, Japan). Each protein sample in 2% (*w*/*v*) SDS/ 4.5 M urea/ 2% (*v*/*v*) glycerol/25 μg/mL (*v*/*w*) bromophenol blue was incubated for 15 min at 60 °C and separated usingSDS-polyacrylamide gel electrophoresis (PAGE) using 12% gels, followed by western blot analysis using the semi-dry transfer apparatus (Bio Craft, Tokyo, Japan). To avoid nonspecific binding of the antibody to a nitrocellulose membrane, the membrane was soaked in 0.1 M Tris-HCl pH 7.6, 0.15 M NaCl, 0.1% Tween 20 (TNT) containing 5% (*w*/*v*) skim milk (blocking buffer) overnight at 4 °C. The first antibodies, O-17 (100 ng/mL, IBL, Gunma, Japan), CD9 (25 ng/mL MM2/57, Bio-Rad, CA, USA), or anti-actin (1:10,000 dilution; A5316, Sigma-Aldrich Co. LLC, St. Louis, MO, USA), were diluted in the blocking buffer and were reacted with the membrane for 60 min at room temperature. After washing the membrane with TNT, the second antibodies diluted in the blocking buffer were reacted for 60 min at room temperature. Bound primary antibodies were reacted with horseradish peroxidase-conjugated goat anti-rabbit IgG or anti-mouse IgG (1:10,000 dilutions; Cell Signaling Technology) secondary antibodies and were visualized using an Immobilon^®^ Forte Western membrane substrate (Merck KGA, Darmstadt, Germany) according to the manufacturer’s protocol. The signals were detected using the ChemiDoc Touch MP system with Image Lab 5.2.1 software (Bio-Rad). 

### 4.6. Effect of MMP-9 and Caspase Inhibitor on OPN Synthesis and Gene Expression in LPS-Stimulated PMA-Differentiated THP-1 Macrophage

PMA-differentiated THP-1 macrophage which was made as described above was cultured with LPS with and without protease inhibitors. MMP-9 inhibitor (5 μM, Santa Cruz, TX, USA) or caspase-8 inhibitor (10 μM, Z-IETD-FMK, Selleck Chemicals, TX, USA) were used at non cytotocic concentrations as described previously [23,24]. After 24 h of culture, supernatants were obtained. Obtained cell lysates were used for western blot assay as described above (4.5). Total RNA was extracted with an SV Total RNA isolation system according to the manufacturer’s protocol (Promega, WI, USA). mRNA transcription of OPN variants was analysed using reverse transcription PCR. Complementary DNA was synthesized by reverse transcription PCR using 3 μg total RNA and the Superscript III First Strand Synthesis System (Liftechnologies, Co., Carlsbad, CA, USA). One microliter of complementary DNA was used for standard PCR to amplify the *OPN* transcripts (OPNa, 361 bp; OPNb,319 bp; OPNc, 281 bp; OPN-4, 238) with the primers og0052 (5′-ACTACCATGAGAATTGCAGT-3′) and og053 (5′-TGGTGAGAATCATCAGTGTC-3′). The PCR reaction mixture, in a final volume of 25 μL, contained 1× PrimeSTAR buffer (Mg^2+^ plus) (Takara Bio, Inc., Shiga, Japan), 0.4 μL dNTP mixture solution (0.2 mM each dNTP), 0.2 μM each of primers, and 0.625 unit PrimeSTAR HS DNA polymerase. Amplification was carried out with the first denaturation at 98 °C for 10 s followed by 30 cycles of denaturation at 98 °C for 10 s, annealing at 60 °C for 10 s, extension at 72 °C for 30 s, and the final extension at 72 °C for 5 min. The amplicon was subjected to electrophoresis in a 1% agarose gel that included EtBr.

### 4.7. Statistical Analyses

One-way repeated measures ANOVA (multiple comparisons) was used to compare the results of ELISA, and the results of exosome distribution and particle mean size. *p* values < 0.05 were considered statistically significant. All statistical analyses were performed using GraphPad Prism software, version 7 (GraphPad Software Inc., San Diego, CA, USA).

## 5. Conclusions

FL and cleaved forms of OPN were identified in the Exo isolated from the culture supernatants from LPS-stimulated PMA-differentiated THP-1 macrophages. The Exo were identified using qNano multiple analyzer and western blotting using the CD9 antibody. The amount of OPN in the culture supernatants that was present in the Exo was 14–19%. These findings suggest that LPS stimulation of macrophages facilitates the encapsulation and release of the Exo containing OPN. Thus, the released Exo is a suitable vehicle to transfer OPN to the target cells.

## Figures and Tables

**Figure 1 ijms-21-08490-f001:**
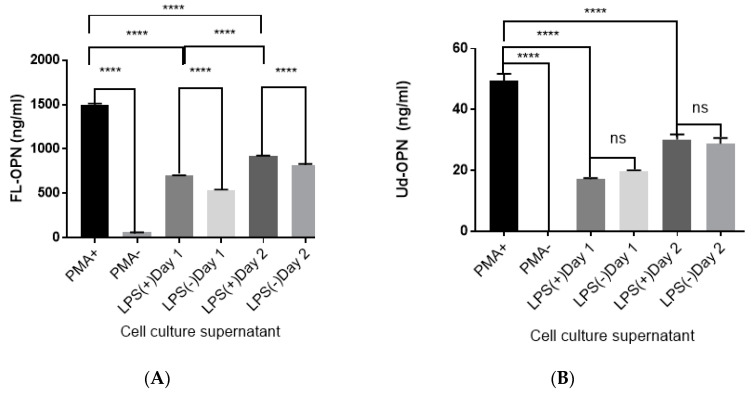
Amount of full-length osteopontin FL-OPN (**A**) and Ud-OPN (**B**) in the culture supernatants of stimulated cells. PMA+, PMA-; THP-1 cells stimulated in the presence or absence of PMA. LPS+, LPS-; PMA-differentiated THP-1 macrophages were stimulated cultured in the presence or absence of LP. LPS-stimulated samples were obtained after culture for one (day 1) and 2 (day 2) days. FL-OPN and Ud-OPN were measured using an ELISA kit (JP27158, IBL, Gunma, Japan) and Human OPN DuoSet ELISA Development System Kit (R&D Systems, Minneapolis, MN, USA), respectively. Data are the means and standard errors of the mean. **** *p* < 0.0001, ns not significant. Representative data of three independent experiments are shown.

**Figure 2 ijms-21-08490-f002:**
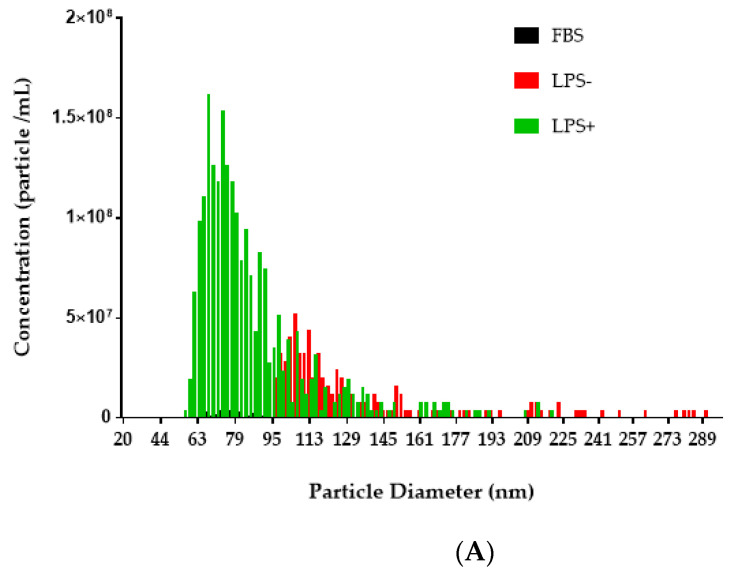
TRPS analysis of the isolated exosomes. The Exo were isolated using the EXO-Prep Exo precipitation kit from the culture supernatants and were analyzed using Nanopore (NP100, Izon). Size distribution (**A**), concentration (**B**), and particle size (**C**) were shown. A representative data of three independent experiments were shown. FBS; medium containing 10% Exo-free fetal bovine serum. LPS (+); LPS (−); PMA-differentiated THP-1 macrophages stimulated with and without LPS for 24 h. **** *p* < 0.0001, ns not significant.

**Figure 3 ijms-21-08490-f003:**
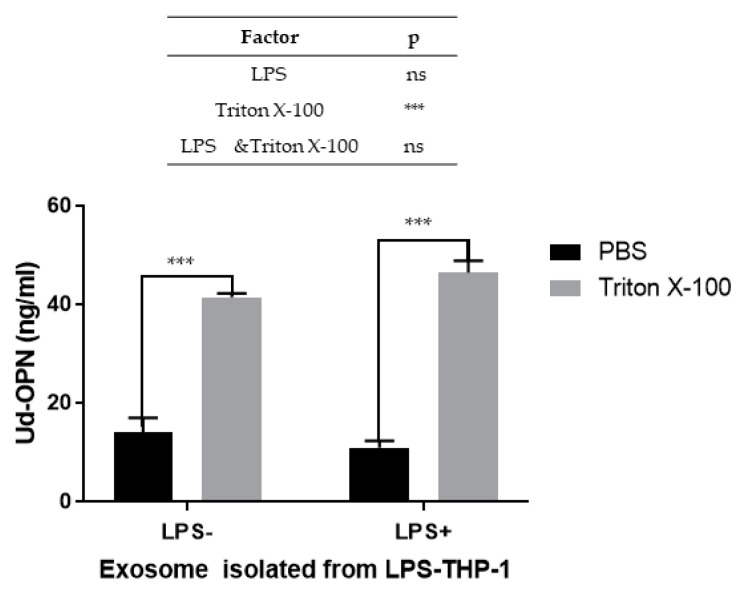
Effect of Triton X-100 on ELISA (Ud-OPN) using Exo isolated from culture supernatants. LPS-; non-stimulated PMA-differentiated THP-1 macrophages LPS+; LPS-stimulated PMA-differentiated THP-1 macrophages. Either phosphate-buffered saline (PBS) or PBS containing Triton X-100 (1%) were used as diluents in the ELISAs. *** *p* < 0.001: a significant difference, ns: no difference; analysed using GraphPad Prism 7 software Representative data of three independent experiments are shown.

**Figure 4 ijms-21-08490-f004:**
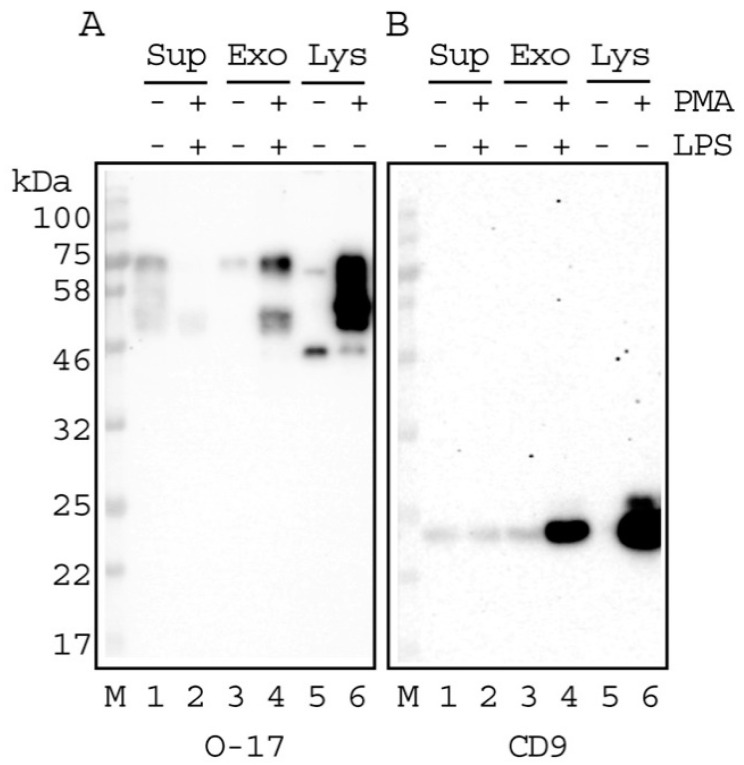
Western blot analysis of OPN in the Exo released from 48 h LPS-stimulated PMA-differentiated THP-1 macrophages; O-17 (**A**) and CD9 (**B**) antibodies were used. Ten micrograms of concentrated supernatant (Sup) and the isolated Exo and 3 μg of cell lysates (Lys) were subjected to SDS-PAGE (12%) followed by immunostaining. LPS(−)_THP-1(+); culture of the PMA-differentiated THP-1 macrophages. LPS(+)_PMA(+); culture of the LPS-stimulated PMA-differentiated THP-1 macrophage; LPS(−)_PMA(−); culture of the non-stimulated cells. LPS(−)_PMA(+), culture of PMA-differentiated THP-1 macrophage. M, molecular weight marker.

**Table 1 ijms-21-08490-t001:** Comparisons of ratios of Ud-OPN/FL-OPN in culture supernatants (CS) and the Exo.

	PMA		LPS _Day1		LPS _Day2	
The volume of CS ^#^ (μL)	9000					
The volume of Exo ^$^ (μL)	900					
	FL-OPN	Ud-OPN	FL-OPN	Ud-OPN	FL-OPN	Ud-OPN
Concentration in CS (ng/mL)	1496	49.6	697	17.1	924	29.3
Total protein in CS (ng)	13,462	446	6270	154	8316	263
Concentration in Exo (ng/mL)	102	12.9	1.024	27.0	1.558	55.5
Total protein in Exo (ng)	91.8	11.7	922	24.3	1402	50.0
Ratios of total protein Exo/CS (%)	0.68	2.61	14.7	15.8	16.9	19.0
Ratios of Ud/FL in CS (%)	3.31		2.46		3.17	
Ratios of Ud/FL in Exo (%)	12.7		2.63		3.56	

^#^ CS: cell culture supernatant. ^$^ Exo: exosome. Representative data of three independent experiments are shown.

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
