# Peer review of "Stimulation of THP-1 Macrophages with LPS Increased the Production of Osteopontin-Encapsulating Exosome"

_ijms, 2020, doi:10.3390/ijms21228490_

Round 1
Reviewer 1 Report
The authors have addressed most of the concerns I had with the previous review. The new experimental data clearly add to the contribution of this paper. Thus, I recommend an accept without changes.
Reviewer 2 Report
The authors have satisfied this reviewer's concerns.
This manuscript is a resubmission of an earlier submission. The following is a list of the peer review reports and author responses from that submission.
Round 1
Reviewer 1 Report
The present study reports the presence of both full-length and cleaved OPN in the exosome, this finding could provide a basis for better understanding of LPS-mediated pro-inflammatory pathway, hence could be useful for related diseases’ monitoring and treatment. However, some issues remain to be solved in order for the study to be published in this journal.
Major comments:
- In result section, the subtitle should be changed into the statement including main finding rather than methods.
- To determine possible involvement of intracellular proteases, such as MMP-9 and caspase-8, the authors can check the Exo/supernatant OPN full length and cleaved form after the inhibition of the abovementioned protease activities using selective inhibitors.
- Figure 2 showed clear size distribution of the exosome; however, the significance of this result has not been explained thoroughly. Is that indicating fragmentation of large Exo into smaller one by LPS stimulation? The author should provide more explanation for the importance of this data.
- In the Abstract, authors described as follows: “Ud/FL ratios became low after LPS-stimulation, indicating the stimulation of FL-OPN synthesis by LPS.” However, in the Discussion, the authors explained “The lower ratios of Ud-/FL-OPN in the Exo from the LPS-stimulated PMA-differentiated THP-1 macrophages may reflect lower cleavage activity of LPS-stimulated cells.” Which one is your conclusive speculation?
- Why didn’t you include FL-OPN data in Fig. 3?
- Some term is confusing. “functional N-terminal OPN fragment (truncated OPN)”, “cleaved form of OPN (N-half OPN)”, “Ud-OPN”. These are all the same. Need to unify.
- Is there any possibility that existence of both the full and cleaved OPN within cell is due to an alternative splicing rather than proteolytic cleavage?
- Figure legends must include detailed experimental conditions.
1) The authors didn’t include LPS concentrations in anywhere.
2) Regarding Fig. 2, How long did you treat cells with LPS?
Minor comments:
- In Figure 1, include meaning of ****.
- In Table 2, check the symbols or letters like @l @l
- Unify the units - mL vs ml, hours vs h, minutes vs min
Lines 79-80, delete the space between “of” and “PMA-“
Line 172, the highest induced protein -> the most highly induced protein
Line 218, “4.3.1 Exo-Prep one step Exosome Isolation” should be deleted.
Line 224, 300 g -> 300 x g 10,000g -> 10,000 x g
Line 232, 4.4 -> 4.3
Line 235, “and” must be deleted.
Line 240, 4.5 ELISAs -> 4.4 ELISA
Line 250, 4.6 -> 4.5
Line 266, 4.7 -> 4.6
Line 274, exo -> Exo
Reviewer 2 Report
The subject of the manuscript entitled “Exosomes derived from LPS-stimulated THP-1 cells 2 contain full-length and cleaved forms of osteopontin” is related to the characterization of the calcium-binding glycophosphoprotein Osteopontin (OPN)-full-length (FL-OPN) and cleaved (Ud-OPN) form-in exosomes (Exo) isolated from lipopolysaccharide (LPS)-stimulated and non-stimulated THP-1 cells differentiated to macrophages. The Exo isolated through Exoprep or ultracentrifugation were identified by qNano 18 multiple analyzer and western blotting with a CD9 antibody.
Based on the complex investigation, the authors suggest that the synthesis of full length OPN in macrophages is facilitated by LPS stimulation and that the protein is subsequently cleaved in cells or Exo by proteases. The result indicate the possible role of Exo as suitable vehicle to transfer OPN to target cells.
The manuscript is well written and organized and fits with the scope of the journal.
I have only a minor comment-to improve the quality of the Figure 4.
Reviewer 3 Report
The authors detected OPN in exosomes. On lines 160 and 161 they state this is the first time OPN has been detected in exosomes (which would be very interesting), but then on lines 189-191 site a manuscript describing OPN detection in exosomes. The data reported here appears to be a start for a mature study, but in its present form lacks depth.
Overall, this is a very difficult manuscript to read. It is very difficult to understand what the authors are saying due to excessive jargon, and each paragraph requires multiple readings gain understanding. On lines 258-260 I cannot understand what is being said.
The authors discuss possible physiological roles for the different forms of OPN, but the data is purely descriptive and not convincing. Figure 2 is not necessary. What does this have to do with the physiology? Figure 3 is not necessary. Who cares about the improvement of a procedure unless it has implications for the physiology? The two Western blots shown are not very convincing., and what primary antibody was used for OPN detection and at what concentration?
There is no detail on how the cultures were performed. How many cells? What were the actual treatments (how much of the treatment was added)? How many times were the experiments repeated?
using the term "undefined" OPN is not very reassuring.